# GFLAgent: Green Federated Learning Agent for Alleviating Heterogeneity

## Abstract

Federated Learning (FL), as a privacy-preserving distributed machine learning paradigm, faces significant challenges in terms of data and device heterogeneity in practical applications. In this paper, we present a novel Large Language Model Agent decision system, called Green Federated Learning Agent (GFLAgent), for alleviating the challenges arising from data and device heterogeneity within the FL tasks. GFLAgent is efficient and energy friendly, and meets the requirements of green computing. GFLAgent dynamically monitors the status of each client, selects and reasonably allocates them to different layers to achieve efficient asynchronous training, and responds to unexpected situations during training. Furthermore, to optimize overall system expenditure, we implement a strategy that minimizes local training overhead and the updates costs for clients with historically subpar performance. The experimental results show that GFLAgent outperforms SOTA methods and can be quickly ported to other distributed machine learning frameworks to improve efficiency.

## 1 Introduction

Distributed machine learning has improved the efficiency of artificial intelligence model training, but it has also exposed the issue of customer data privacy. Participants from different institutions tend not to transmit private data when participating in training tasks. For this reason, Federated Learning (FL) McMahan et al. (2017)is proposed. FL is crafted to aggregate local model updates from distributed devices without centralizing data, thereby safeguarding privacy and ensuring robust model training through iterative global refinement. Nonetheless, FL can incur significant computational, network, and performance expenses during the training phase. This trade-off may stem from the inherent architectural features Lo et al. (2022) of FL and the challenges posed by statistical heterogeneity Luo et al. (2022) across the distributed datasets.

The key challenge in FL is statistical heterogeneity, where data is frequently imbalanced and non-identically independently distributed (non-IID) Zhao et al. (2018). Thus, the indiscriminate training of all clients and the subsequent model updates, without considering the distribution and quality of data, can degrade model performance. Given these challenges, a pertinent question arises: is it possible to devise a model that selects clients in a more scientific and effective manner for each training round? This question is central to the evolving field of client selection algorithms in FL.

There have been many attempts to address heterogeneity issues. In the early days, many studies emphasized the issue of lagging behind, which was caused by uneven distribution of computational data and computing power. To solve this problem, Tier-based Federated Learning System (TiFL) Chai et al. (2020a) utilize a multi-layer mechanism to mitigate the impact of stragglers. However, it calls for well-designed layer to avoid excessive loss of model information. Besides, it may also have a bias problem that tends to choose faster layers. Federated learning method with Asynchronous Tiers (FedAT) Chai et al. (2020b) has taken a step further by adopting a novel weighted aggregation heuristic algorithm, assigning higher weights to slower layers to prevent the preferences.

Nevertheless, all these traditional scheduling optimization requires engineers to spend a lot of time to design the weights and optimize models. The Large Language Models (LLM) have shown the potential in improving current work Zhao et al. (2023) since it has experienced explosive growth in the past two years and is proven to make amazing progress in generative tasks. Based on LLMs, we use LLM-based agent Wang et al. (2024) to improve automatic logical reasoning.

What's also noteworthy is that current algorithms ignore the situation that the unexpected disconnection of a client with other tasks could result in anomalies within a particular tier, which can negatively influence the communication efficiency.

To avoid these unexpected situations impacting model performance and time of communication, we build a buffer zone to involve abnormal clients during training, storing their updates while uploading selectively. What's more, compared with elaborate strategy of clients allocation in different tiers and model weight, GFLAgent use LLM-based agent to adjust the server's decision automatically with less parameter engineering. The Agent uses a carefully designed method to evaluate the actual contribution of clients to overall performance, selects some clients to participate in training, improves task efficiency, and reduces energy costs. These aspects demonstrate that our methods align with Green FL, promoting sustainable environmental development Kim & Saad (2024).

To sum up, our contributions are as follows:

- We have designed a FL system for alleviating the heterogeneity of data and devices in distributed learning, which we name GFLAgent. Within this system, we have constructed an efficient and robust asynchronous federated learning framework, integrated with an LLM-based Agent serving as the scheduler for the entire system. This scheduler is also readily transferable to other FL systems.

- Innovatively addressing potential issues within the Tiers framework, we propose a buffer designed specifically for outlier clients operating within the Tiers. Accompanying this buffer is a vigilant monitor, capable of swiftly identifying these outlier clients and relocating them into the buffer. This mechanism significantly enhances the robustness of the heterogeneous FL system.

- We conducted experiments on standard datasets and compared them with other advanced algorithms. The results demonstrate that our method outperforms existing methods significantly. Compared to the FL algorithm with full clients participation, our method maintains similar accuracy. Besides, in some cases of heterogeneous data distribution, our model performs better than other SOTA methods.

## 2 RELATED WORKS

### 2.1 FEDERATED LEARNING

In recent times, Federated Learning (FL) has emerged to help protect information security by enabling model training without the need for central data storage, whose protocol requires the selected clients to update their model using local data, while asking the server to aggregate updates from the clients to make the model better Nishio & Yonetani (2019).

Several classic FL algorithms, including Federated Averaging (FedAvg) McMahan et al. (2017), FedProx Tian et al. (2018), and Stochastic Controlled Averaging for Federated Learning (SCAFFOLD) Karimireddy et al. (2020), have been introduced to enhance the quality of FL. FedAvg leverages local stochastic gradient descent (SGD) Goodfellow et al. (2016) on each client, with a server performing model averaging. This method is robust against unbalanced and non-IID data distributions and effectively reduces the number of communication rounds McMahan et al. (2017).

FedProx expands on FedAvg by tackling statistical heterogeneity among clients, thus enhancing convergence behavior in realistic, heterogeneous networks McMahan et al. (2017). Algorithms like SCAFFOLD further optimize FedAvg by employing control variates to counteract client-drift in updates, a strategy known as variance reduction Karimireddy et al. (2020).

Given that each client may handle varying amounts of data, this discrepancy can affect subsequent communications and the quality of the updated model Nishio & Yonetani (2019). Moreover, these variations can profoundly influence model accuracy, convergence rate, and fairness across clients Fu et al. (2023).

Addressing both system heterogeneity (variations in hardware configurations among clients) and statistical heterogeneity (differences in data distributions among clients) is essential to overcoming similar challenges in FL McMahan et al. (2017). Client sampling and selection are pivotal in solv-

ing these problems. Here are two main strategies: client classification for improved asynchronous training and bolstering communication efficiency between clients and servers.

Asynchronous training is a response to the delays caused by stragglers. Extensive research Stich (2018) Li et al. (2019b) Karimireddy et al. (2020) Yang et al. (2021) has concentrated on reducing their impact in distributed networks to avoid performance deterioration. However, the outright exclusion of stragglers might result in the loss of critical data needed for model enhancement. Hybrid Federated Learning (HFL) Truex et al. (2019), which incorporates techniques such as Asynchronous Decentralized SGD (AD-SGD) Lian et al. (2018), mitigates this by integrating delayed local updates into the central model Li et al. (2021).

An alternative approach to managing straggler clients involves a synchronous intra-tier model updating strategy. Yet, this method may induce biases due to its favoring certain clients. In contrast, Federated learning method with Asynchronous Tiers (FedAT) improves upon Tier-based Federated Learning System (TiFL) by combining synchronous intra-tier training with asynchronous cross-tier training, effectively reducing the straggler effect and enhancing both convergence speed and test accuracy.

Another pioneering method focuses on minimizing communication frequency in FL by selectively choosing clients based on model update thresholdsRibero & Vikalo (2020). This strategy, inspired by Ornstein-Uhlenbeck processes and SGD Mandt et al. (2016), ensures that only updates that surpass a certain significance are conveyed to the server, thereby streamlining communication efficiency.

We have also seen some other advanced models. FedBalancer Shin et al. (2022) is a case in point, which exemplifies a notable model that refines time-to-accuracy performance by harmonizing client-server interactions. It selects samples with substantial statistical utility and dynamically forecasts optimal deadlines for each training round, contingent on the variability of client training data, which in turn optimizes the learning trajectory.

These advancements underscore the ongoing efforts to refine FL methodologies, balancing the complexities of client heterogeneity with the need for efficient and effective model training across distributed environments.

## 2.2 LARGE LANGUAGE MODELS AGENTS

The rise of large language models (LLMs) Radford et al. (2019) has been a cross-age change in AI, with GPT-3 Dale (2021) from OpenAI being a standout example. These models use a transformer architecture and are trained on huge amounts of text, making them great at producing text that feels like it was written by a person Kasneci et al. (2023). However, scaling up model size alone has not proved effective for achieving greater performance on tasks such as arithmetic, commonsense, and symbolic reasoning Rae et al. (2021).

To help LLMs handle complex reasoning, researchers have come up with new ideas like "Chain of Thought prompting" (CoT) and "Tree of Thought" (ToT). Unlike expensive rationale-augmented trainingLing et al. (2017) and fine-tuning methodsCobbe et al. (2021), CoT Wei et al. (2022) leverages the <input, chain of thought, output>template, enabling LLMs to perform few-shot prompting for reasoning tasks efficiently. While ToT Yao et al. (2024) allows LLMs to self-assess and choose between advancing or backtracking along different reasoning paths during global decision-making. Furthermore,Graph of Thought (GoT) Besta et al. (2024) generalizes CoT and ToT to more intricate thought patterns, enhancing reasoning without the need for model updates.

The progression to LLM agents signifies a transformative leap in AI's capability for complex reasoning and task execution. LLMs can be applied to more than just conversation-based language tasks. Researchers are now looking at how LLM agents can be used in real-life situations to solve complex problems. For example, they are working on improving reasoning in games like Werewolf Xu et al. (2023) and in AI environments like Smallville Park et al. (2023), as well as developing new ways for agents to think and make decisions.

By applying the strengths of LLMs and techniques like CoT and ToT, we aim to make progress in how LLM-based agents tackle real-world challenges.

## 3 PROBLEM STATEMENT

Suppose we have a federation of *m* clients engaged in a collaborative training endeavor, each possessing a unique dataset that is not to be disclosed to other clients or the central server. Our primary aim, akin to the foundational goals of federated learning, is the optimization of the global objective function to ensure the model's performance is minimized in a collective sense.

$$\min_{\{w,i\}} F_{\mathcal{G}}(w) = \frac{1}{m} \sum_{i=1}^{m} F_i(w) \tag{1}$$

Given the distributed nature of machine learning, it is imperative to account for the overall time cost associated with the tasks. During training, the time expenditure for each round may initially be denoted as $t_{\mathcal{G}}^r = \max\left(t_1^r, t_2^r \ldots, t_m^r\right)$, representing the global perspective. Upon the implementation of client selection strategies, such as those aimed at optimizing the process, the time cost could be further refined to $t_{|\mathcal{S}|}^r = \max\left([t_i^r \text{ in } |\mathcal{S}|]\right)$. Here, $|\mathcal{S}|$ symbolizes the time cost incurred in the *r*-th round for the subset of clients that have been selected.

$$\min_{\{i\}} T = \sum_{r=1}^{R} t^r \tag{2}$$

Ultimately, in a distributed system, we also aim to minimize energy expenditure while meeting the task requirements. The term $\tau_i$ represents the average computational power of device *i*, which signifies the operational duration of the *i*-th client during the *r*-th round.

$$\min_{\{F_{\mathcal{S}} < \hat{x}, i\}} C = \sum_{i=1}^{m} \sum_{r=1}^{R} \tau_i \cdot t_i^r \tag{3}$$

Additionally, the server's time cost and energy consumption are linearly related to the number of computation rounds and the number of clients participating in the submission and aggregation tasks per round. Although this aspect is relatively minor compared to the training costs incurred by the clients and is relatively straightforward, it will not be elaborated upon extensively here. However, it will be taken into account and compared in the experimental section.

## 4 PROPOSED SCHEME

### 4.1 FRAMEWORK

Initially, we define the common practice problem encountered by FL as having statistical heterogeneity. Therefore, following by FedProx, we set the basic model update strategy $h_k(\cdot)$ for the client as follows:

$$h_i(w_i) = F(w_i) + \frac{\lambda}{2} ||w_i - w||^2 \tag{4}$$

There is no denying the excellence of employing a tiered asynchronous design approach. In the realm of federated learning, traditional resource scheduling techniques like task offloading and load balancing are inapplicable due to the non-shareability of client data. Efforts to offload non-sensitive computational tasks from slower clients to cloud servers, as seen in prior research, were attempts to navigate these challenges within the federated learning paradigm.

Our comprehensive design draws inspiration from the tiered asynchronous learning model, but we have identified potential issues that arise when there is a significant variance in client data volume or device capabilities. Such disparities could necessitate the creation of numerous tiers, each with substantial waiting times. The unexpected disconnection or preoccupation of a client with other tasks could lead to anomalies within a particular tier. To mitigate this, our design minimizes the number of tiers and incorporates buffers to manage outlier clients effectively.

The client selection mechanism is a cornerstone of our approach. Historically, such mechanisms were governed by rule-based algorithms that had to account for time expenditure, computational efficiency, and communication overhead, evaluating clients' historical performance to determine their

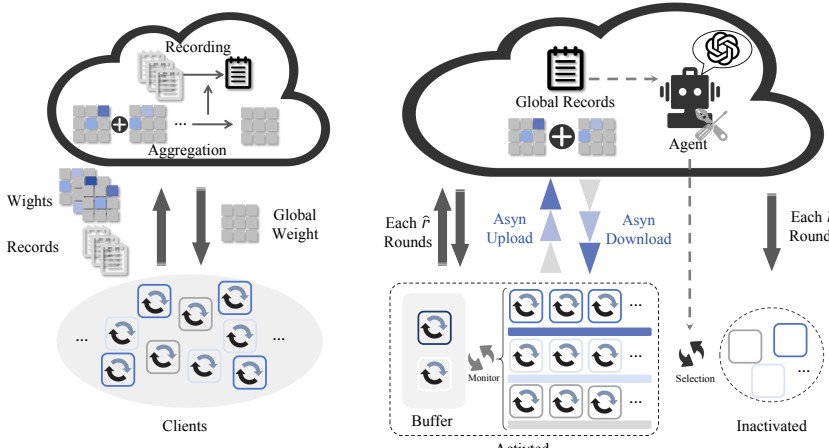

Figure 1: The diagram on the left represents the update process using the naive FedAvg algorithm, whereas the diagram on the right showcases the comprehensive framework of our innovative GFLAgent system. Initially, the task is updated using the method depicted on the left. After several iterations, we transition to the framework on the right, where the GFLAgent selects clients and allocates them to suitable tiers for processing. Once the tasks within each tier are completed by the clients, the weights are uploaded to the server for aggregation. Upon successful aggregation, the server disseminates the updated model to all clients engaged in computations across various tiers. Meanwhile, clients operating in the buffer, as well as those in the active client pool, will only receive the most recent aggregated model during the subsequent selection cycle. The GFLAgent's client selection is primarily guided by the global operational data retrieved from or provided by the server.

participation in subsequent tasks. This process was highly experience-dependent and demanded extensive hyperparameter tuning. To streamline this, we have developed an automated Selection Agent, GFLAgent, leveraging Large Language Models (LLMs). GFLAgent utilizes historical training data to determine which clients should be involved in the next round and assigns them to the appropriate tiers. It also integrates data processing tools to supply the LLMs with more coherent operational data.

In summary, our framework encompasses three main components: (1) a tiered federated learning system with defined update strategy rules (2) buffers and strategies to safeguard outlier clients within the tiers, and (3) a decision-making Agent for client selection and allocation in federated learning, powered by LLMs. For detailed insights, one can refer to the framework's pseudo-code and Fig 3.

## 4.2 BUFFER FOR OUTLIER

Regardless of how client selection is optimized, the naive single-column parallel structure will inevitably generate waiting time for laggards due to differences in data volume and computing power. The method of setting up different levels can alleviate this issue; however, the updates within each level are always problematic. For instance, in earlier studies, the experimental settings had a large time redundancy for each level, and the number of levels was fixed. In our scheme, we only set a default of three levels to accommodate these data, and we have designed a buffer to accommodate outliers that are selected, adopting different update strategies.

Firstly, we introduce different Tiers to enable asynchronous updates in federated learning. Assuming we set up $K$ layers to accommodate selected activations, considering that layers with fast update speeds may have bias effects on the entire model due to more submissions over a period of time. we introduce model bias compensation weights due to differences in update speed. Optimized aggregation method shows in 5. In the equation, $T_{tier_k}$ represents the time cost by $Tier_k$, and $max(T_{tier})$ represents the longest time cost among all $Tiers$.

$$F(w) = \sum_{k=1}^{K} \frac{T_{tier_k}}{max(T_{tier})} h_i(w_i)_{\{tier_k, i \in ||s||\}} \tag{5}$$

---

**Algorithm 1** GFLAgent Workflow

---

**Input**: $m$ clients with their respective datasets $\mathcal{D}_i$, inital global model wight $w$, $R$: the global iteration round. $\hat{R}$: the round of introducing GFLAgent

**Output**: Finished global model

1: **for** round $r = \hat{R}, ..., R$ **do**
2:     Run FedAvg algorithm and write records
3: **end for**
4: **for** round $r = \hat{R}, ..., R$ **do**
5:     **for** each $\hat{r}$ rounds **do**
6:         Agent selects clients and distribute to $Tiers$
7:         $Tiers$ **parallel do**:
8:           Clients train in $\mathcal{D}_i$
9:           Monitor move abnormal client i to $Buffer$
10:          Send $w_i$ to server until all train done
11:          Server aggregate $w_i$ send back to client $w$
12:         Clients in buffer train and keep best $w_i$
13:         Server aggregate $w_i$ send back to **all** client $w$
14:     **end for**
15:     Server write the latest information to records
16: **end for**
17: **return** finished model weight $w$

---

**Algorithm 2** Buffer Monitor Algorithm

---

**Input**: $m$ clients with their respective datasets $\mathcal{D}_i$, initial global model wight $w$, $R$: the global iteration round. $\hat{R}$: the round of introducing GFLAgent

**Output**: Client in Tiers and Buffer

1: **if** r at client selection round **then**
2:     $t_{max}, t_{min} \leftarrow$ Agent decision
3: **end if**
4: **for** Client in $Tiers$ **do**
5:     **if** Client training time $t_i > t_{max}$ or $t_i < t_{min}$ **then**
6:         Move client to Buffer
7:     **end if**
8:     **for** Each round all client in Tier finished training **do**
9:         Use $Monitor(\cdot)$ by Equation 7.
10:     **end for**
11: **end for**

---

Using the above method may not be reasonable, due to ignore clients in the system that have a large amount of high-quality data and run quickly. Simply compensating for weights based on time is unfair to them. Therefore, we introduce a data contribution parameter to optimize this weight. The improved formula is as follows, in which $\triangle\ell$ represent the difference between the loss function and the local update before and after the previous round. $Norm(\cdot)$ is the normalize operator, which normalizes all the parameters in this round. $w_i, w$ respectively represent the weight parameters of client i and the model after the previous iteration aggregation.

$$Norm(\frac{\triangle\ell}{log(1 + \frac{\lambda}{2}||w_i - w||^2)}) \sum_{k=1}^{K} \frac{T_{tier_k}}{max(T_{tier})} \underset{\{tier_k, i \in ||s||\}}{h_i(w_i)} \tag{6}$$

Specifically, clients are selected based on their historical performance to participate in subsequent training rounds. After being selected, they are allocated to different levels according to their historical speed. Previous reseaches assumed that the performance of clients would remain consistent, they would fully meet the time expectations designed for different tier. However, issues such as device occupation and communication failures could lead to sudden slowdowns that might recover within

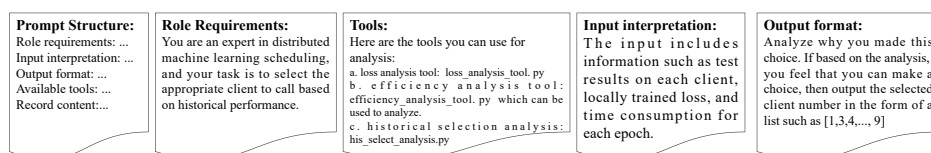

Figure 2: Prompt and Content

a certain period. Or the device may suddenly go offline during training and not recover. The detail shows in Algorithm 2.

The following formula shows the Monitor evaluation whether or not move the client to Buffer from Tier, where $\sigma^2(T_{tier})$ is the variance of the running time in this tier, and $\bar{T}_{tier}$ is the average training time per round. This formula has been improved based on the variance contribution ratio method. Default relaxation factor default relaxation factor $\varphi = 1$ and referring to Li et al. (2019a), to ensure system stability, Monitor only employed for the time-cost **top 20%** or **bottom 20%** of clients. Monitor will move client to buffer if $Monitor(\cdot) > \varphi$.

$$Monitor(\cdot) = \frac{\left(t_i - \bar{T}_{tier}\right)^2}{\sigma^2(T_{tier}) \cdot (n-1)} \tag{7}$$

The design of this buffer is immediate; all selected outliers are placed in this buffer. We do not need to particularly consider the updates within the buffer and the external updates. The buffer is isolated from the external environment until the redistribution round arrives. Clients in the buffer complete updates independently, and each round of updates is stored within the buffer. If the update is useful, the buffer stores this update, a mechanism that ensures the updates within the buffer remain optimal. Finally, at the set redistribution round, the buffer will upload the optimal model parameters of each client to the server for aggregation. It should be noted that if a client in the buffer recovers to meet the expected speed of a certain level after several updates, the client will be released from the buffer to the appropriate level with the optimal model parameters.

### 4.3 LLMs based Agent for Efficiency Improvement

In contrast to traditional methods, which often rely on mathematical techniques to meticulously craft evaluation functions or rules, our approach involves scoring the performance of all clients after a comprehensive run using these functions. The selection process is refined by setting thresholds or selecting a certain percentage of clients for participation. Recognizing the potential loss of data or computational resources that may occur when some clients are discarded, we employ a random supplementation strategy to include those initially overlooked. For example, FedBalance incorporates this strategy by merging selected and unselected data into a parallel data processing mode. To transcend the limitations of manual configuration, we have integrated an LLM-based Agent as an embedded decision-making tool to assess client participation and adjust the server's overall hyperparameters dynamically, allowing for tailored adjustments to the training process at various stages. The architecture we've designed, as depicted in the figure, is built on a server with a high baseline computational capacity to optimize the system's overall performance.

#### 4.3.1 Basic Scheme.

To minimize the learning curve, we've adopted a straightforward approach using prompt words for task management. The beauty of this method lies in its simplicity; by tweaking the prompt templates and key elements, one can effectively manage task scheduling. This process is accessible to engineers with basic knowledge, enabling them to perform system optimizations that yield immediate, observable results. The rationale behind these adjustments is also transparent, as the outputs from LLMs provide clear insights into the decision-making process.

#### 4.3.2 AGENT-BASED SCHEME.

In the past two years, there has been a proliferation of concepts and schemes based on Large Language Models (LLMs) Agents. Agents endowed with tools, reflection, and memory have become the mainstream design paradigm. In response to this, we have crafted an Agent optimized for federated learning.

- **Memory:** In our framework, memory is segmented into short-term and long-term facets. While short-term memory retains recent operations and dialogues, our primary focus lies with long-term memory, which documents execution adjustments. The term "other" indicates a comprehensive archive of task records post-execution. Our method involves extracting and analyzing these records to generate metadata that captures participant involvement, client outcomes, temporal engagement, and accuracy. This dataset further explains client selection patterns and the decision logic of LLMs across successive rounds.

- **Tools:** LLMs have demonstrated that their capacity for mathematical reasoning is not as robust as initially anticipated. To address this, we have designed a suite of tools to augment LLMs, compensating for their deficiencies in computational aspects. These tools consist of text transformation utilities, key information extraction, data processing capabilities, and foundational machine learning model scripts. The design of Contribution Evaluation Module with analysis is demonstrated in Appendix A.

- **Chain of Thought:** The implementation of CoT in this scenario essentially follows a prompt template to completion. This allows LLMs to assess whether the requirements are met and to output templates that more closely align with the demands. Drawing inspiration from the React method, we have constructed the CoT for our scenario, which we will illustrate through a set of indicative words and processes that represent our CoT framework.

## 5 EXPERIMENTS

### 5.1 SETUP

#### 5.1.1 INFRASTRUCTURE.

Our experimental environment is anchored in servers deployed with the Linux Ubuntu 22.04 Server operating system. The server is powered by two Intel Xeon Platinum 8352V CPUs, complemented by 256 GB of RAM. It is further endowed with 8X NVIDIA RTX3090 GPUs, facilitating the capability to perform heterogeneous simulations that adeptly harness the combined strengths of CPU and GPU resources.

The federated learning framework employed in our experiments is derived from PFLlib Zhang et al. (2023), a code library that serves as a robust platform for investigating federated learning and personalized federated learning scenarios. The machine learning models engaged in our experimental tasks, including Deep Neural Networks (DNNs), Convolutional Neural Networks (CNNs), and the ResNet-18 He et al. (2016), leverage PyTorch as their underlying architecture, ensuring a high degree of flexibility and performance. The default training batch size for each client is set to 10, with each client training only one epoch locally in each round

#### 5.1.2 BASELINE MODELS.

In our investigation, we have incorporated a suite of recent baseline models related to client selection and the robust efficiency of federated learning for comparative analysis. These include FedAT, TiFL, and FedBalancer, with FedAvg also integrated as a comparative benchmark. Additionally, we will contrast these models against several regular client selection strategies to evaluate their performance comprehensively.

#### 5.1.3 STATISTICAL HETEROGENEITY.

In our experiments, we have configured a variety of data heterogeneity challenges of varying difficulty to comprehensively assess the performance of our proposed methods. Regarding independent and IID data, we have simply established two types of distributions where each client possesses

approximately the same or varying amounts of data. Additionally, we have employed two widely-used settings for statistical heterogeneity: the pathological setting and the practical setting. In the pathological setting, each client receives a fraction of the total number of classes in the dataset, such as one-fifth for the MNIST dataset LeCun et al. (2010), which has 10 classes. Consequently, while the entire FL task is conducted across the full dataset, each client is allocated data corresponding to only two classes. For the practical setting, we have assigned the data distribution across clients following a Dirichlet distribution Lin et al. (2020), which is the default configuration.

## 5.2 PERFORMANCE

In this section, we conducted accuracy assessment experiments for all selected models. The experimental setup involved 20 participating clients. As demonstrated in Table 1, our experiments utilized a time-constraint mode, given that we strategically selected a subset of clients for training; had we adopted the same number of global rounds for training, our model's performance would likely be inferior to that of models trained with the full complement of clients. Additionally, we compared our results with those obtained by FedAvg after 1000 full rounds of training (denoted as FedAvg-F). Please note, the average rate at which GFLAgent completes 1000 global rounds is $5.7\times$ faster than that of FedAvg-F.

Table 1: The test results compare the accuracy (%) of all algorithms under the condition of GFLAgent completing 1000 training rounds. Notably, FedAvg-F represents the outcome of FedAvg's 1000 full training rounds without time constraints. In Practical setting, $\beta = 0.05$ is employed as the Dirichlet distribution parameter.

| Settings | IID | Pathological | | | Practical | | |
|---|---|---|---|---|---|---|---|
| **Dataset** | MNIST | MNIST | Cifar-10 | Cifar-100 | Cifar-10 | HAR | AG News |
| FedAvg-Full | 97.63±0.45 | 93.79±0.23 | 54.10±0.33 | 24.15±0.40 | 59.71±0.42 | 81.17±0.38 | 78.72±0.27 |
| FedAvg | 96.10±0.17 | 91.99±0.42 | 49.50±0.29 | 16.01±0.45 | 51.97±0.22 | 77.52±0.45 | 75.39±0.31 |
| TiFL | 96.21±0.24 | 89.71±0.37 | 50.32±0.28 | 17.55±0.43 | 52.67±0.34 | 78.62±0.30 | 77.21±0.41 |
| FedAT | 97.21±0.41 | 92.12±0.26 | 52.21±0.35 | 19.28±0.25 | 56.41±0.31 | 78.20±0.29 | 77.82±0.36 |
| FedBalancer | 96.05±0.27 | 92.61±0.14 | 52.07±0.08 | 21.97±0.10 | 55.67±0.12 | 79.15±0.20 | 78.12±0.25 |
| **GFLAgent** | 97.58±0.32 | 93.75±0.39 | 55.75±0.27 | 23.25±0.42 | 57.51±0.37 | 81.07±0.23 | 77.95±0.38 |

## 5.3 ROBUSTNESS

### 5.3.1 DATA HETEROGENEITY

As shown in Table 2 Robustness to data heterogeneity is also a key focus of our attention. In this section, we have adjusted the parameters of the Dirichlet distribution, specifically setting $\beta = 0.1, 0.3, 0.7$ for the purpose of simulating data heterogeneity.

Table 2: On this experiment, the framework we proposed demonstrates the highest efficiency, using the same benchmark as before, which is GFLAgent operating over 1000 global rounds. GFLAgent(0.7T) represents the performance of GFLAgent at 0.7 times as the general time. At the same time, we also used 50% accuracy as a cross-section to study the time (in seconds) for different methods to achieve this accuracy on all datasets three times.

| Cifar-10 | $\beta = 0.1$ | $\beta = 0.3$ | $\beta = 0.7$ | Cifar-10@Acc50% | [5,5,5,5] | [10,0,10,0] | [8,2,2,8] |
|---|---|---|---|---|---|---|---|
| FedAvg-F | 64.33±0.15 | 67.39±0.11 | 69.97±0.12 | FedAvg | 2789 | 1439 | 2910 |
| FedAvg | 48.70±0.41 | 63.64±0.31 | 65.96±0.12 | FedAvg-1T | 2931 | 1557 | 3021 |
| TiFL | 59.08±0.27 | 65.12±0.19 | 67.88±0.22 | TiFL | 771 | 720 | 1028 |
| FedAT | 60.43±0.20 | 66.82±0.20 | 69.08±0.33 | FedAT | 788 | 656 | 802 |
| FedBalance | 61.02±0.30 | 66.76±0.23 | 69.33±0.41 | FedBalance | 2107 | 1051 | 2759 |
| FedProto | 58.24±0.15 | 60.16±0.32 | 63.63±0.31 | FedProto | 3872 | 1957 | 3315 |
| **GFLAgent(0.7T)** | 60.94±0.26 | 66.39±0.31 | 68.65±0.29 | - | | | |
| **GFLAgent** | 62.14±0.21 | 67.20±0.20 | 69.25±0.18 | **GFLAgent** | 591 | 488 | 789 |

### 5.3.2 DEVICE HETEROGENEITY

We have also conducted relevant tests on the robustness of heterogeneous devices and the occurrence of errors on the client side in the system. The tests have shown that our solution also performs well in addressing such issues. We have set different device heterogeneity and abnormal situations: for heterogeneous device situations, we use [GPU, $\frac{1}{2}$GPU, CPU, $\frac{1}{2}$CPU] to represent the allocation of heterogeneous devices, and [10,0,10,0] represents a situation with 10 GPU devices, 0 half-performance CPU device. Details are shown in the right part of Tabel 2

### 5.3.3 ABNORMAL STATE

Consider that each device will have $\zeta\%$ performance fluctuations or training delay anomalies. In addition, we also discussed the delay time caused by offline or long-term client downtime. Other methods do not have the ability to handle this situation. Due to the buffer and monitoring mechanism we designed, GFLAgent can quickly handle such faults. The results indicate that GFLAgent's ability to handle occasional delay situations is quantitatively **one order of magnitude** higher. Details are illustrated in Appendix Table 5

### 5.4 EFFICIENCY PERFORMANCE

The client selection mechanism not only enhances performance in terms of data heterogeneity but, more importantly, it conserves energy. We conducted a straightforward energy consumption test, which was a cross-sectional study. All clients run on the same device, and our training efficiency has improved by **37.6%** compared to FedAvg. Compared to its other best model FedAT (3 Tiers), it has improved by **8.9%**. The training energy consumption is **21.4%** lower than FedAvg, and **10.2%** higher than its other best model FedBalancer-S. Details in Appendix Figure 3

### 5.5 ABLATION EXPERIMENT

We conducted ablation experiments with the default baseline set at 3.4.1, which mainly includes three aspects of ablation: buffer handling of abnormal situations, agent construction (as set in 3.4.2), and contribution calculation module for overall efficiency improvement. We tested the impact of the proposed different modules on the overall method. The w/o Agent module represents that there is only one large language model and prompt word template as a simple scheduler, but both the buffer and contribution evaluation modules are available, and we introduced the usage of the latter in the prompt word template. It can be seen that the buffer module is designed to prevent delay exceptions and does not have a significant impact on accuracy.

Table 3: Results for Ablation Experiment

| Time = 200s | Accuracy(%) |
| --- | --- |
| GFLAgent | 54.15 |
| w/o Tier Buffer | 53.29 |
| w/o Agent Module | 50.91 |
| w/o Contribution Evaluation | 51.78 |
| FedAvg | 45.56 |

## 6 CONCLUSION

To address the heterogeneity challenges inherent in federated learning and to boost the efficiency of such systems, we have introduced a framework known as GFLAgent. This methodology leverages a tiered federated learning strategy and incorporates a buffer mechanism to address the outlier scenarios that may arise among clients within tiers. Moreover, we have pioneered the use of an LLMs-based Agent, tailored for client selection in federated learning, to function as the orchestrator of the system. Our experiments have substantiated the efficacy of our approach. Importantly, the framework we have developed is not only versatile but also capable of enhancing the efficiency of existing federated learning frameworks when integrated with them.

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

## A    APPENDIX A: CONTRIBUTION EVALUATION ANALYSIS

The decrease of local loss which shows that the model performs well on local datasets doesn't naturally guarantee the good performance of global model because of the inconsistent data distribution. We will prove this in later paragraph.

Fedbalancer Shin et al. (2022) capitalizes on the local loss magnitude during client training to select key contributors. Additionally, to counteract the potential for neglect, it assigns a chance for clients who have previously been overlooked by the efficiency selection algorithm to be included. However, discrepancies in data distribution between local and global datasets may lead to inflated local loss values. Influenced by GPFL Na et al. (2024), we have designed a new selection metric. Our method takes into account the information contained in the loss, as well as the role of the client within the global context. In the calculation, we can approximate the distance between individual and overall data distributions based on differential fuzziness as follows:

$$Dist_i = ||\sum_{j \in \mathbb{N}} \nabla f\left(\boldsymbol{w}_j\right) - \nabla f\left(\boldsymbol{w}_i\right)||_2^2 \tag{8}$$

The smaller the $Dist_i$, the closer the direction between local and global aggregation updates. Then, we can analyze the loss. It's easy to draw a conclusion that the decrease of loss combined with the increase of accuracy promises the data distribution consistency.

To further analyze the problem, we have the following assumptions:

**Assumption 1:** In each round, agent $i$ has a quality $\mu_i$ drawn from a distribution with mean $\mu_i$ and variance $\sigma_i^2$. $\mu_i$ contains two parameters, $acc_i$ and $loss_i$ respectively.

**Assumption 2:** Since we only need to select clients in a whole, it's fine to draw a rough conclusion without fine-tuning complex parameters.

Under the above assumptions, we get 9.

$$Cont = (acc_i^{t+1} - acc_i^t) \cdot (loss_i^{t-1} - loss_i^t) \cdot log(1 + \frac{1}{Dist_i}) \tag{9}$$

We are going to illustrate the reasonability of this equation. *Cont* is a controllable item, determining clients used for updates. $acc_i^{t+1} - acc_i^t$ and $loss_i^{t-1} - loss_i^t$ represent the accuracy and loss difference between two continue updates. These two should be positive to ensure the improvement of model.

We use $log(1 + \frac{1}{Dist_i})$ to adjust clients' contributions to ensure the global convergence and stability of the model. Specifically, when $Dist_i$ is large, meaning that the client's data distribution differs significantly from the global data distribution, $\frac{1}{Dist_i}$ will be small, which in turn makes $log(1 + \frac{1}{Dist_i})$ also small. This reduces the contribution of that client's update to the global model update.

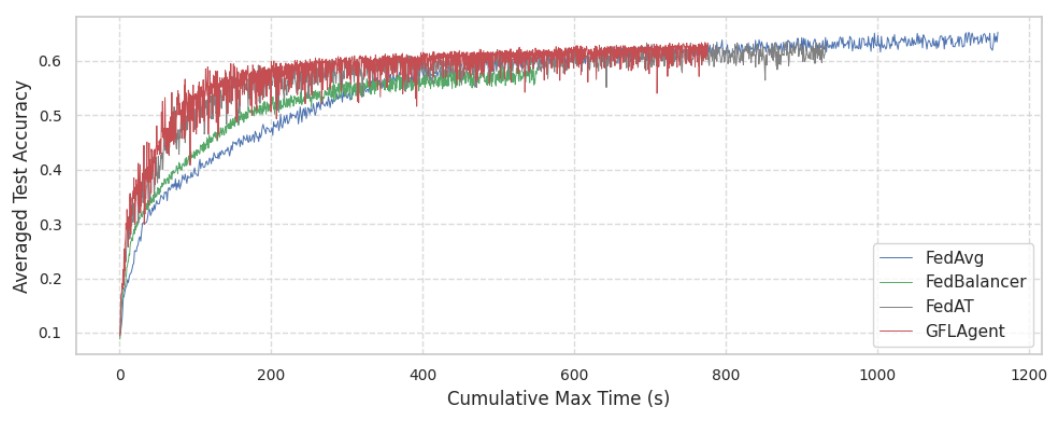

Figure 3: Comparison of Training Efficiency

# B   APPENDIX B: SUPPLEMENTARY EXPERIMENTAL RESULTS

We describe the impact of different modules on method efficiency in Fig 4 , and the impact of different modules on energy overhead in Fig 5.  At the same time, we also provided a Prompt template that we used in B.5

## B.1   PERFORMANCE COMPARISON OF DIFFERENT BASE LLMs

We conducted relevant tests on different pedestal models before conducting all experiments.  The results indicate that the input length of qwen-14b is only 2k tokens, and with prompt words, it can only accommodate about 3 rounds of training records (with approximately 12-20 clients participating). For performance evaluation, we simply used the time to achieve the same training accuracy as a comparison metric. Qwen-14b, as a scheduler, is not considered due to the significant difference in distance between multiple experiments. All times in the table represent the average time it takes to generate usable results under the maximum input context (although the results are not simply a list containing client numbers, we use text rules to match and extract a list of content that matches our experiment).  In the end, we chose Moonshot as the LLM base model for this experiment.  The result detail in

Table 4: Results of Performance Comparison.

| Model | The longest read round | Average Performance | Time Consumption Per Call (s) |
|---|---|---|---|
| Qwen-14B Local | 3 | - | 14.6 |
| GLM-4-9B-chat-1m Local | 650 | 100% | 200.7 |
| moonshot-v1-32k | 42 | 117% | 28.1 |
| GLM-4-Long | 650 | 119% | 175.6 |
| Qwen-Max-128k | 85 | 112% | 72.9 |

## B.2   SUPPLEMENTS OF EFFICIENCY EXPERIMENT

In the efficiency test, we set a global deadline of 1000 iterations to compare the accuracy performance of different methods at the same time scale.  The results in 3 show that our method can achieve higher accuracy in the same time, and the final result of our method is only slightly lower than FedAvg with full client participation. After meeting 1000 rounds, Fedbalancer stops first, followed by GFLAgent, then FedAT with slightly lower performance than our method, and finally FedAvg.

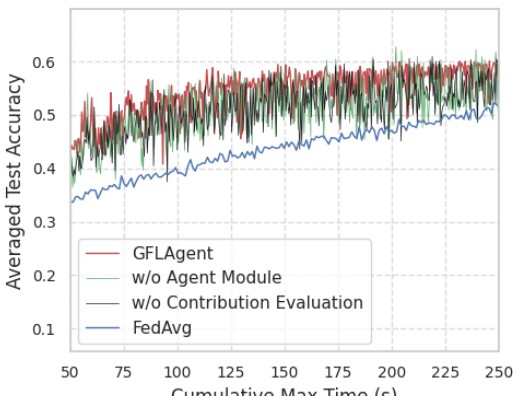

Figure 4: Ablation Experiment for Efficiency.

## B.3 SUPPLEMENTS OF ABNORMAL STATE

Each client has a probability of experiencing a delay failure of 10 seconds, and the following time is the delay compared to the average completion time without failure. Due to delayed failures, FedBalancer deadline estimation will increase, leading to further increase in overall time costs. In our experiment, other methods were unable to handle offline and long-term downtime failures. The unit of all delay time is second.

Table 5: Results for Abnormal State.

| Method | $\zeta=0.1\%$ | $\zeta\%=1\%$ | $\zeta\%=5\%$ | Processing Delay |
|---|---|---|---|---|
| FedAvg | 196.27 | 1758.9 | 6932.88 | - |
| FedAT | 127.81 | 867.9 | 2376.25 | - |
| FedBalancer | 278.51 | 1891.24 | 5416.52 | - |
| GFLAgent | 28.87 | 216.06 | 614.78 | 0.53 |

## B.4 SUPPLEMENTS FOR ABLATION EXPERIMENT

From the time accuracy description figure 4, it can be seen that thanks to the client selection and stratification strategy of the basic LLM, GFLAgent and other ablation methods performed better than the baseline FedAvg. Due to the hierarchical aggregation scheme, the faster layer method has a significantly faster iteration time per round than FedAvg.

It can be seen in Figure 5 that although there is not much difference in performance improvement when GFLAgent and other modules are ablated, this may be because LLM has already evaluated the energy training loss performance in response and made choices that are more in line with green computing

## B.5 PROMPT EXAMPLE

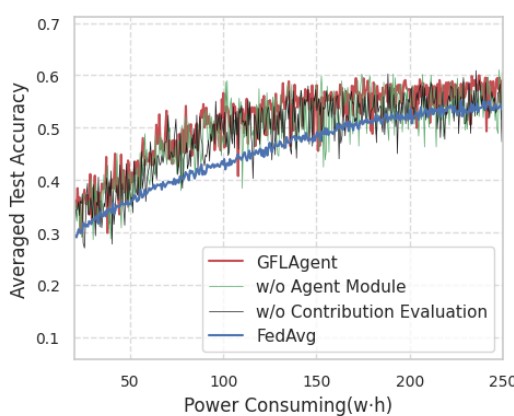

Figure 5: Ablation Experiment for Power Consuming.

---

**Algorithm 3** An Example for Prompt of Agent Construction

---

1: Role: {You are a distributed machine learning expert who can choose by analyzing training records}
2: Input instruction: {You can read the training records, the most basic training records are: {round - number of rounds, testing accuracy on all participating clients in that round, performance of participating clients {client number, local epoch, corresponding epoch loss, time consumed per epoch} and overall training}}
3: Output requirement: {It is possible to output an analysis, but the final result that needs to be output is a list []. If you feel that all the information is sufficient to output, then simply follow the required list}
4: Tools: {Your available tools include analysis tools, which you can use to analyze existing training evaluation logs and combine analysis, include 'a.py', 'b.py'}
5: History: {} (History records are mainly used to build agents to reflect on previous actions and generate more appropriate answers)
6: Training log: {} (read from file)

---

