# OpenReview forum: "GFLAgent: Green Federated Learning Agent for Alleviating Heterogeneity"
_ICLR.cc/2025/Conference — Submitted to ICLR 2025_

### Official Review · Reviewer_J62R · 2024-10-22

**Soundness:** 2
**Presentation:** 2
**Contribution:** 2
**Rating:** 3
**Confidence:** 5

**Summary:**

Comment 1:
In Section 4, the paper introduces a buffer system to handle outlier clients, but how exactly does the system determine when a client should be classified as an outlier? More details on this process, including criteria for moving clients in and out of the buffer, would be helpful to avoid any negative impact on the overall system performance.

Comment 2:
The paper talks about using weights to correct bias in the tiered updates. What specific factors are taken into account when calculating these weights, particularly in situations where clients have very different data sizes or capabilities?

Comment 3:
In Section 4.3, the method for selecting clients based on historical performance is introduced, but how does the agent prioritize certain clients over others? Does the agent use specific metrics or rules to determine which clients are most valuable for future rounds of training?

Comment 4:
In Section 5, the results on statistical heterogeneity could be expanded. Would it be possible to provide more detailed comparisons across datasets to give a clearer picture of how the system handles different non-IID data conditions?

Comment 5:
In Section 5, how does device heterogeneity, particularly when clients have varying levels of computational power, affect the overall accuracy and communication efficiency? Considering this is a common issue in practical federated learning deployments.

Comment 6:
Does the buffer mechanism account for long-term client downtimes, and if so, how does it ensure that these clients do not negatively impact the model’s convergence and accuracy?

Comment 7:
The energy efficiency improvement claim is interesting but lacks depth. Could the paper provide more specific evidence on how GFLAgent reduces energy consumption compared to other models, especially across different hardware setups?

**Strengths:**

1. The paper introduces a creative use of Large Language Models (LLMs) to automatically select clients in federated learning, making the process more efficient by reducing manual work. This approach helps optimize training without needing constant adjustment by engineers.

2. The use of a buffer to manage slow or problematic clients is a strong point. It helps ensure that clients with issues don’t slow down the whole process, improving the reliability of the system.

3. The paper presents a variety of experiments using different datasets, showing that the method works well even when the data is distributed unevenly across clients.

**Weaknesses:**

1. While the LLM-based client selection method is a good idea, the paper doesn’t explain enough about how the system decides which clients to choose.

2. The paper mentions a buffer system for handling outliers but doesn’t give enough detail about how clients are moved into or out of the buffer.

3. While the paper talks about device heterogeneity, it doesn’t provide enough details on how different devices with varying computational power affect the system’s performance.

**Questions:**

1. How does the LLM-based agent update its decisions during training? Does it keep learning and improving as training progresses, or does it rely only on past client performance data?

2. What happens if a client stays offline for a long time? How does this affect the model’s accuracy and overall performance, especially in real-world scenarios?

3. How much does each part of the system (like the buffer and the LLM-based agent) contribute to reducing energy use? Can the paper provide more detailed data on energy savings for each part?

---

### Official Review · Reviewer_EAU4 · 2024-10-27

**Soundness:** 3
**Presentation:** 2
**Contribution:** 2
**Rating:** 3
**Confidence:** 3

**Summary:**

- This study introduces GFLAgent for considering heterogeneity in federated learning, with a focus on green computing
- The unique aspect, I believe, is the use of LLMAgent for device selection (tiering) in federated learning

**Strengths:**

- The consideration of green computing, including resource efficiency, electricity, and carbon dioxide reduction, is important
- The paper effectively illustrates its design components

**Weaknesses:**

- Aside from the use of the LLM-based agent, some other ideas have appeared in recent studies (e.g., high-level buffer concept in EcoFed). Can you clarify the unique contribution of this study?
- Although the title mentions “green” computing, the main design focuses on data and resource heterogeneity, similar to other studies. The primary aim and purpose of the study are somewhat unclear.
- The novel aspect of this work seems to be the LLM-based agent. However, the explanation of it in the main content is brief.
- The explanation of the evaluation is confusing. Some references are incorrect, and the interpretation of the results is incomplete. For instance, in Section 5.3.2, I cannot determine which values support the statements. If this refers to the right side of Table 2, why is data heterogeneity evaluated by accuracy, but device heterogeneity by time?
- There is no evaluation of the green computing aspect in the main although the paper title includes it

**Questions:**

Including the weakness above, I raise the following questions

- How are K,  t_min, and  t_max  determined, and how does the evaluation change with different values? Some empirical values are referenced from other studies, but such values might need to be adjusted depending on the system
- The models used in the evaluation are not clearly described. The authors mention DNNs, CNNs, and ResNet-18, but it’s unclear how the numbers in the tables and figures are derived. Which models are they based on?
- I believe the comparison techniques are not the most state-of-the-art. Consider checking a survey (e.g., Heterogeneous Federated Learning: State-of-the-art and Research Challenges, ACM Survey)
- (minor) Table 2 is not mentioned in the text. Perhaps the sentence referencing Table 3 should also mention Table 2

---

### Official Review · Reviewer_fBgq · 2024-10-29

**Soundness:** 1
**Presentation:** 1
**Contribution:** 2
**Rating:** 3
**Confidence:** 4

**Summary:**

This paper proposes GFLAgent, an LLM-based agent to adjust the server’s decision automatically with less parameter engineering. GFLAgent uses a carefully designed method to evaluate the actual contribution of clients to overall performance, selects some clients to participate in training, improves task efficiency, and reduces energy costs. Additionally, GFLAgent builds a buffer zone to involve abnormal clients during training, storing their updates while uploading selectively.

**Strengths:**

1.	Compared with elaborate strategy of clients allocation in different tiers and model weight, LLM-based agent attains better performance.
2.	Both the data and device heterogeneity are considered in this paper.

**Weaknesses:**

1.	Research motivation could be improved. It’s not clear how “unexpected disconnection” affects the performance of FL. The authors should provide more details about the “unexpected disconnection” problem.
2.	The description about workflow of GFLAgent (in Section 4) is hard to understand. It would be better to present the GFLAgent in a more clear and detailed manner. Also, in line 252, the “Fig.3” seems to be a misquote as it does not contain details of GFLAgent workflow.
3.	In FL environments that do not include abnormal clients (which is common in the real world), does GFLAgent adversely affect FL performance?
4.	The baselines used in the experiment is not representative enough of the SOTA approach. It may be better to include more fresh and relevant work in the comparison experiments.

**Questions:**

In FL environments that do not include abnormal clients (which is common in the real world), does GFLAgent adversely affect FL performance?

---

> ### Author Response · Authors · 2024-11-28
>
> Indeed, your considerations are comprehensive. In our architecture, LLM will be frequently called, which will be an overhead. At the same time, it will consume time from reading training log files to selecting output. Our framework is designed as an asynchronous system. That is to say, LLM analysis may be initiated in the K-th round, and the results may be obtained in the K+m round, but the client used during this period still adopts the K-round strategy. The latest strategy will only be executed in the next update round after receiving the completed strategy. This process is asynchronous and wait free, so it can be seen as the scheduler running independently of the client, and even server aggregation.
> In addition, we will strive to improve the comparison with other SOTA methods and also increase discussions on issues related to participant anomalies.

---

### Official Review · Reviewer_BpqS · 2024-10-31

**Soundness:** 2
**Presentation:** 1
**Contribution:** 2
**Rating:** 5
**Confidence:** 3

**Summary:**

This paper introduces GFLAgent, which uses an LLM-based agent on the FL-server side to schedule the federated learning process to tackle the general data and computation heterogeneity concerns. The authors provide experiments on heterogeneously partitioned datasets to compare GFLAgent with popular FL strategies to shows its advantages in trained model performance and efficiency.

**Strengths:**

Using LLM-based scheduler sounds like an interesting idea for orchestrating the FL process, compared with traditional rule-based scheduling, as it can provide an explanation for the schedule decisions at the same time.

**Weaknesses:**

Though this paper presents a great and interesting idea to use LLM-based agent to schedule the FL process, the paper's related work, problem statement, and experiment details have space to improve, specficically:

(1) The introduction section (Section 1) mentions tier-based asynchronous FL algorithms such as TiFL requires human-designed tiers, however, the authors should also mentions its advantages compared with the following recent work in ICLR 2024 which generates tiers automatically based on history computation time.

Li, Zilinghan, Pranshu Chaturvedi, Shilan He, Han Chen, Gagandeep Singh, Volodymyr Kindratenko, Eliu A. Huerta, Kibaek Kim, and Ravi Madduri. "FedCompass: Efficient Cross-Silo Federated Learning on Heterogeneous Client Devices Using a Computing Power-Aware Scheduler." In *The Twelfth International Conference on Learning Representations*.

(2) The related work section (Section 2) talk about asynchronous federated learning, and the paper should cite the following two popular asynchronous algorithms.

Xie, Cong, Sanmi Koyejo, and Indranil Gupta. "Asynchronous federated optimization." *arXiv preprint arXiv:1903.03934* (2019).

Nguyen, John, Kshitiz Malik, Hongyuan Zhan, Ashkan Yousefpour, Mike Rabbat, Mani Malek, and Dzmitry Huba. "Federated learning with buffered asynchronous aggregation." In *International Conference on Artificial Intelligence and Statistics*, pp. 3581-3607. PMLR, 2022.

(3) The problem statement section (Section 3) should be made more clear, for example:

- There is no definition for F in equation 1
- It is unclear why equation 1 is min_{w,i} instead of min_{w}
- For equation 2, it is unclear why it is min_{i} but i is not in the euqation

(4) Figure 1 is supposed to be the main description of the GFLAgent algorithm, but it is never referred in the main text, same for Figure 2.

(5) From lines 205 to 207, the authors wrote: "Efforts to offload non-sensitive computational tasks from slower clients to cloud servers, as seen in prior research, were attempts to
navigate these challenges within the federated learning paradigm." I wonder if there is any reference to support this statement.

(6) The Experiment section should be improved the clarity, for example:

- In line 464, the author mentions "As shown in Table 3…" - I wonder how Table 3 contributes to the following statements about data heterogeneity.
- In line 489, the author mentions "Details are shown in the right part of Table 3" - I think it is supposed to be Table 2.
- It is still confusing to me what [10, 0, 10, 0] means.
- As the LLM-based agent is considered the most important key part of this paper, it is important to clearly state what LLM is used for the algorithm in the main paper, instead of the Appendix.

(7) There are few minor problems as well:

- line 150 contains some strange characters
- In line 561, the citation to Deep Learning, Yoshua Bengio appears twice.
- The subtitles for Appendix B are all together, which makes it very hard to read.

**Questions:**

Nothing more than what i mentioned above

---

> ### Author Response · Authors · 2024-11-28
>
> Thank you for reviewing our manuscript and providing thoughtful comments. We have made some corrections in the newly submitted version. Your comments will also be taken into account in future paper improvements, although some technical routes are currently difficult to complete modifications in a short period of time.

---

> > ### Comment · Reviewer_BpqS · 2024-12-02
> > **Thank you!**
> >
> > Thank you  for your response. I maintain my previous scores for the paper as a good number of my original concerns are still valid.

---

### Official Review · Reviewer_Bndn · 2024-11-03

**Soundness:** 1
**Presentation:** 2
**Contribution:** 1
**Rating:** 3
**Confidence:** 4

**Summary:**

This paper proposes three strategies to enhance federated learning (FL) performance in the presence of client heterogeneity (both statistical and system heterogeneity). The strategies include:  introducing penalty weights for client updates, employing a buffering strategy to address client outliers, and utilizing large language models (LLMs) for client selection/task scheduling decisions. While the objective of improving FL performance in heterogeneous environments is relevant, it is not a novel topic.

**Strengths:**

The combination of multiple techniques to improve FL performance in heterogeneous settings.

**Weaknesses:**

There are many weaknesses:
The first two strategies presented are not new, as similar mechanisms have been explored in existing literature, beyond just the FedAT reference cited. A more comprehensive review of prior work is needed for readers to understand the essential differences between these two strategies and existing approaches.

The proposed techniques rely heavily on established methods, but the paper lacks sufficient clarity in explaining the parameters of and the rationale behind the formulas used. The content should be self-explanatory.

While the paper discusses the use of LLMs for task scheduling and client selection, it does not adequately explain how LLMs arrive at their decisions in this specific context of FL training. Only general background information about LLM is provided.

**Questions:**

Is the decision-making process for LLMs fully automated throughout the federated learning process? What are the specific interactions between the LLM agent and the FL training process?

Please clarify the variables used in equations 5-7.

The authors stated that existing client selection mechanisms account for time, computational efficiency, and communication overhead, and the process is highly experience-based and demanded extensive hyperparameter tuning. Is it case for all existing client selection mechanisms? A broader literature review could be conducted.

---

> ### Author Response · Authors · 2024-11-25
>
> Thank you for your comment. Firstly, regarding the strategy, the basic FedAsync method is the most fundamental asynchronous acceleration method. After our research, we found that FedAT and Fedbalancer are not the latest strategies, but they have shown the most stable performance in performance testing. The reinforcement learning based methods do not provide detailed factual details, and we have not performed well in attempting to reproduce them, making it difficult for their models to converge. Indeed, we acknowledge that there are shortcomings in our recent research on some universal methods, and we will further improve them in the future.
>
> For Questions:
>
> 1.We use LLMs to make decisions for FL clients based on their inclusion of evaluation modes for model training in pre training. Of course, the approach we use is to have it review the changes in loss over time, and this method requires all participating clients to be honest. The selector only needs to read the training logs of each client to make corresponding selection judgments.
>
> 2.The relevant variable representations have been explained in the new version of the main text
>
> 3.Indeed, there are currently methods with strong adaptability, such as FedCompass mentioned by reviewer BpqS, which adaptively performs hierarchical partitioning and client allocation. Thank you for your feedback. We will conduct a more detailed literature review and make improvements in the future.

---

> > ### Comment · Reviewer_Bndn · 2024-11-26
> >
> > Many thanks for authors' response. There are still indeed some issues that need further investigations. I maintain my current scores for the paper.

---

### Meta-Review · Area_Chair_YrX9 · 2024-12-18

**Metareview:**

Reviewers agree on the fact that the paper does not pass the bar for acceptance, and have provided comments that will surely be beneficial to the authors in revising their draft. A short discussion with two reviewers did happen (and short comments from the authors to a third).

**Additional Comments On Reviewer Discussion:**

A short discussion happened with two reviewers.

---

### Decision · Program_Chairs · 2025-01-22

Reject